# High Transcriptional Activity and Diverse Functional Repertoires of Hundreds of Giant Viruses in a Coastal Marine System

Anh D. Ha,[a] (ID) Mohammad Moniruzzaman,[a] (ID) Frank O. Aylward[a]

aDepartment of Biological Sciences, Virginia Tech, Blacksburg, Virginia, USA

**ABSTRACT** Viruses belonging to the *Nucleocytoviricota* phylum are globally distributed and include members with notably large genomes and complex functional repertoires. Recent studies have shown that these viruses are particularly diverse and abundant in marine systems, but the magnitude of actively replicating *Nucleocytoviricota* present in ocean habitats remains unclear. In this study, we compiled a curated database of 2,431 *Nucleocytoviricota* genomes and used it to examine the gene expression of these viruses in a 2.5-day metatranscriptomic time-series from surface waters of the California Current. We identified 145 viral genomes with high levels of gene expression, including 90 *Imitervirales* and 49 *Algavirales* viruses. In addition to recovering high expression of core genes involved in information processing that are commonly expressed during viral infection, we also identified transcripts of diverse viral metabolic genes from pathways such as glycolysis, the TCA cycle, and the pentose phosphate pathway, suggesting that virus-mediated reprogramming of central carbon metabolism is common in oceanic surface waters. Surprisingly, we also identified viral transcripts with homology to actin, myosin, and kinesin domains, suggesting that viruses may use these gene products to manipulate host cytoskeletal dynamics during infection. We performed phylogenetic analysis on the virus-encoded myosin and kinesin proteins, which demonstrated that most belong to deep-branching viral clades, but that others appear to have been acquired from eukaryotes more recently. Our results highlight a remarkable diversity of active *Nucleocytoviricota* in a coastal marine system and underscore the complex functional repertoires expressed by these viruses during infection.

**IMPORTANCE** The discovery of giant viruses has transformed our understanding of viral complexity. Although viruses have traditionally been viewed as filterable infectious agents that lack metabolism, giant viruses can reach sizes rivalling cellular lineages and possess genomes encoding central metabolic processes. Recent studies have shown that giant viruses are widespread in aquatic systems, but the activity of these viruses and the extent to which they reprogram host physiology *in situ* remains unclear. Here, we show that numerous giant viruses consistently express central metabolic enzymes in a coastal marine system, including components of glycolysis, the TCA cycle, and other pathways involved in nutrient homeostasis. Moreover, we found expression of several viral-encoded actin, myosin, and kinesin genes, indicating viral manipulation of the host cytoskeleton during infection. Our study reveals a high activity of giant viruses in a coastal marine system and indicates they are a diverse and underappreciated component of microbial diversity in the ocean.

**KEYWORDS** giant viruses, NCLDV, *Nucleocytoviricota*, viral diversity, kinesin, myosin, cytoskeleton

Large viruses of the phylum *Nucleocytoviricota*, commonly referred to as "giant viruses," are a diverse group of double-stranded DNA viruses (dsDNA) with virion sizes reaching 1.5 µm, comparable to the sizes of many cellular lineages (1–4). A recently

Address correspondence to Frank O. Aylward, faylward@vt.edu.

For a commentary on this article, see https:// doi.org/10.1128/mSystems.00751-21.

proposed taxonomy of this viral phylum delineated 6 orders and 32 families (5), including the previously established families *Asfarviridae*, *Poxviridae*, *Marseilleviridae*, *Iridoviridae*, *Phycodnaviridae*, and *Mimiviridae*. Members of the *Nucleocytoviricota* are known to infect a broad range of eukaryotic hosts; members of the *Poxviridae* and *Iridoviridae* families infect numerous metazoans; members of the *Imitervirales* and *Algavirales* orders infect a wide range of algae and other protists; and members of the *Asfuvirales* infect a mixture of metazoan and protist hosts (6–10). Members of the *Nucleocytoviricota* typically harbor exceptionally large genomes that are often >300 kbp in length, and in some cases as large as 2.5 Mbp (9). Numerous studies have noted the unusually complex genomic repertoires of viruses in this group, which include many genes typically found only in cellular lineages, such as those involved in the TCA cycle (11–13), glycolysis (12), amino acid metabolism (14), light sensing (15, 16), sphingolipid biosynthesis (17, 18), eukaryotic cytoskeleton (19), and fermentation (19). These complex metabolic repertoires are thought to play a role in the manipulation of host physiology during infection, in effect transforming healthy cells into reprogrammed "virocells" that more efficiently produce viral progeny (12, 20, 21).

The genomic complexity of *Nucleocytoviricota*, together with reports that they are abundant in global marine environments, has raised questions regarding the extent to which they manipulate the physiology of their hosts and thereby influence global biogeochemical cycles (22). Cultivated representatives of these viruses are known to infect a broad array of ecologically important eukaryotic algae, including dinoflagellates, haptophytes, brown algae, and chlorophytes, among others (23–26), which suggests they could play important roles in shaping the activity and composition of the microbial eukaryotic community. Even so, our understanding of the distribution and activity of *Nucleocytoviricota* in marine systems has lagged behind that of other viral groups owing to their large size, which precludes their presence in the small size fractions typically analyzed in viral diversity studies (10). Some early studies were prescient in suggesting that these viruses are a diverse and underappreciated component of viral diversity in the ocean (27–30). More recent work has confirmed this view and revealed that a diverse range of *Nucleocytoviricota* inhabit the global ocean and likely contribute to key processes such as algal bloom termination and carbon export (12, 31–36). Some metatranscriptomic studies have also begun to note the presence of *Nucleocytoviricota* transcripts in marine samples, confirming their activity (37–39). Although these studies have vastly expanded our knowledge of the diversity and host range of *Nucleocytoviricota* in the ocean, the activity of these viruses and the extent to which they use cellular metabolic genes they encode during infection remains unclear.

In this study, we constructed a database of 2,436 annotated *Nucleocytoviricota* genomes for the purpose of evaluating the gene expression landscape of these viruses. We then leveraged a previously published metatranscriptome time-series from surface waters of the California Current (39) to assess the daily activity of *Nucleocytoviricota* in this environment. We show that hundreds of these viruses, primarily of the orders *Imitervirales* (including the family *Mimiviridae*) and *Algavirales* (including the families *Phycodnaviridae* and *Prasinoviridae*), were consistently active during this period and frequently expressed genes involved in central carbon metabolism, light harvesting, oxidative stress reduction, and lipid metabolism. Unexpectedly, we also found expression of several viral-encoded cytoskeleton genes, including those that encode the motor proteins myosin and kinesin, and we performed a phylogenetic analysis demonstrating that *Nucleocytoviricota* commonly encode deep-branching enzymes in these protein families. Our findings highlight the surprisingly high activity of *Nucleocytoviricota* in marine systems and suggest they are an underappreciated component of viral diversity in the ocean.

## RESULTS AND DISCUSSION

We examined a metatranscriptomic data set of 16 time points sampled over a 60-h period that were obtained from microbial communities previously reported for the

A)

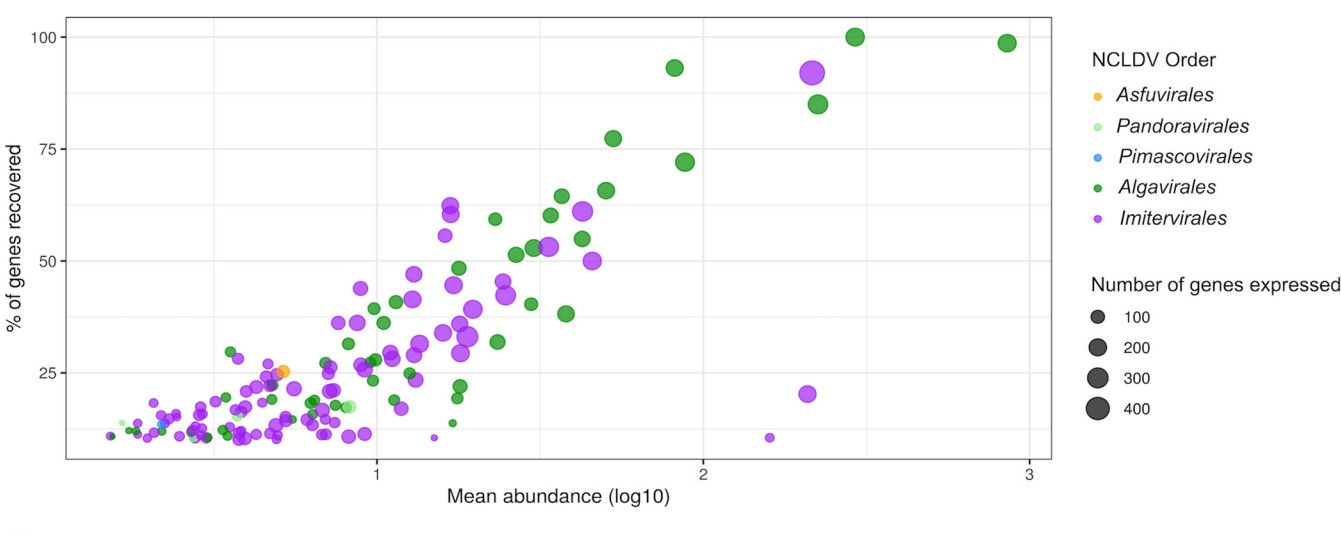

B)

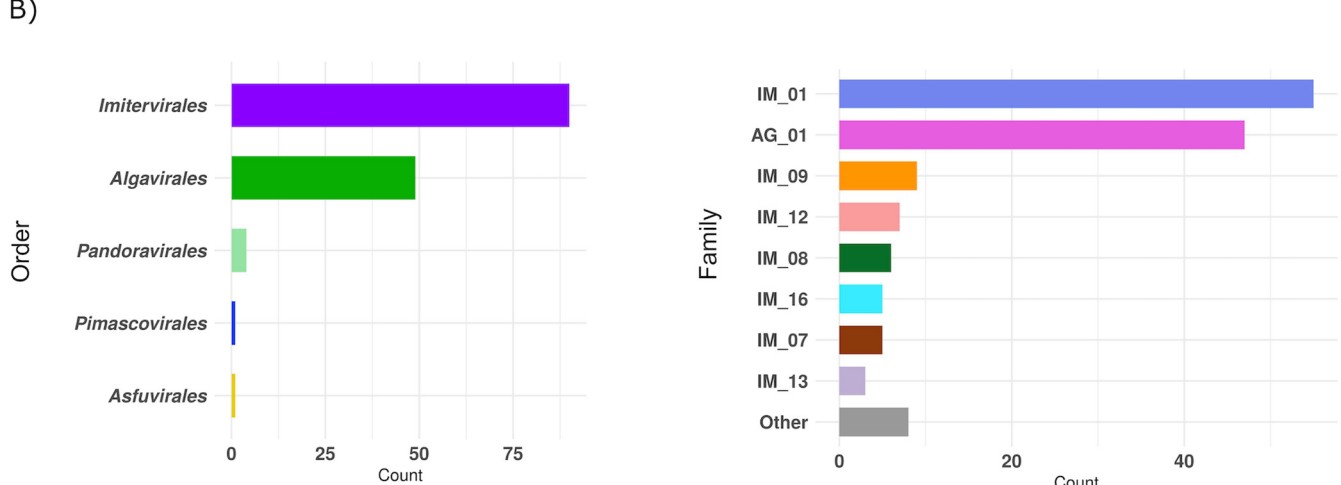

**FIG 1** (A) General statistics of the 145 *Nucleocytoviricota* genomes. Each dot represents a viral genome; the *x* axis shows the mean abundance of the genome across 16 time points (TPM, $\log_{10}$ transformed); the *y* axis shows the percentage of genes in a viral genome that were recovered across all samples; and the dot size is scaled to the total number of genes recovered. (B) The taxonomic distribution of the 145 genomes. The *x* axis represents the total number of viruses in each taxon, and the *y* axis shows the *Nucleocytoviricota* order (left) and family (right). In the family plot (right), families with less than three members are collapsed down into the "Other" category. Abbreviations: IM, *Imitervirales*; AG, *Algavirales*.

California Current system (39). This data set is ideal for examining the activity of marine *Nucleocytoviricota* because it targeted the >5-μm size fraction, which is enriched in many of the eukaryotic plankton that these viruses are known to infect. Altogether, we identified 145 *Nucleocytoviricota* genomes with metatranscriptomic reads mapping over the sampling period, including 90 *Imitervirales*, 49 *Algavirales*, 4 *Pandoravirales*, 1 *Pimascovirales*, and 1 *Asfuvirales* (Fig. 1, see Materials and Methods for details). Expression levels of the 145 genomes with reads mapping are summarized in Data Set S1 in the supplemental material. Of the 145 genomes detected, 9 are complete genomes of cultivated viruses, while 136 are genomes derived from cultivation-independent methods (i.e., metagenome-assembled genomes, or MAGs). Almost half (66) of the 145 *Nucleocytoviricota* genomes have genome or assembly sizes of >300 kbp (Fig. S1), indicating that a large fraction of the active viruses have large genomes with complex functional repertoires. The abundance of each *Nucleocytoviricota* genome in the transcriptomes was positively correlated with the number of genes that could be recovered for that virus (Fig. 1a), which is expected given that only highly expressed

Tree scale: 1 ⊢———————⊣

■ *Imitervirales*
■ *Algavirales*
■ *Pandoravirales*
■ *Pimascovirales*
■ *Asfuvirales*
■ *incertae sedis*

**FIG 2** Phylogeny of *Nucleocytoviricota* with mean read mapping (TPM, natural log transformed) on the outermost bar plot. The order-level classification of *Nucleocytoviricota* is denoted by the branch colors and the outer color strip, and particular families of interest are highlighted in the inner ring. IM_01 corresponds to the *Mesomimiviridae*, IM_16 corresponds to the *Mimiviridae*, and AG_01 corresponds to the *Prasinoviridae*.

viral genes will be recovered for low-abundance viruses in the data set. Over 80% of the predicted genes in five genomes were recovered throughout the sampling period (one *Imitervirales* and four *Algavirales* viruses), indicating that the recovery of most transcripts is possible for some abundant viruses (Fig. 1).

We constructed a multilocus phylogenetic tree of the 145 viruses present in the transcriptomes, together with 1,458 references, so that we could examine the phylogenetic distribution of the active *Nucleocytoviricota* (Fig. 2). The topology of the resulting tree is consistent with a previous tree we constructed using similar methods (12). Given the large diversity of *Imitervirales* and *Algavirales* viruses in the data set, we also analyzed the family level diversity within these orders, using the recently proposed taxonomic framework for the *Nucleocytoviricota* as a guide (5). Of the 145 viruses that were abundant in the metatranscriptomes, the *Mesomimiviridae* (*Imitervirales* family 1) and *Prasinoviridae* (*Algavirales* family 1) were by far the most well-represented (55 and 47 genomes, respectively, Fig. 1B). Interestingly, both of these families also include cultivated representatives that are known to infect marine protists, which suggests possible hosts for the viruses in the transcriptomes. The *Mesomimiviridae* contains haptophyte viruses that infect the genera *Phaeocystis* and *Chrysochromulina* (40–42), while the *Prasinoviridae* contains Prasinoviruses that are known to infect *Ostreococcus*, *Bathycoccus*, and *Micromonas* (8). Transcripts mapping to isolate *Ostreococcus* viruses within the *Prasinoviridae* were previously reported in the original analysis of these

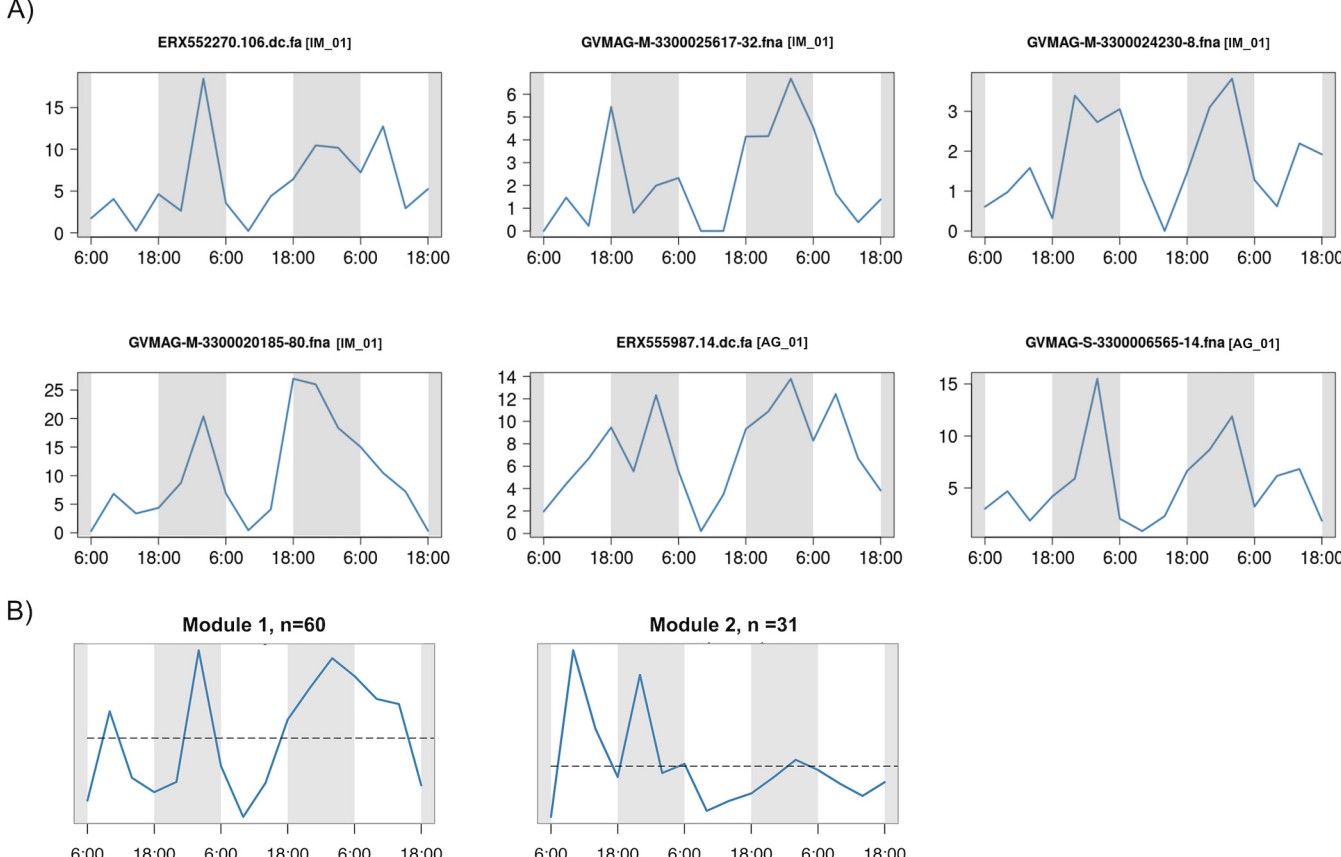

**FIG 3** Expression levels of *Nucleocytoviricota* during the 16 time points studied. (A) Whole-genome expression profiles of six genomes with significant difference in expression level between daytime and nighttime (Mann-Whitney U test $P < 0.1$). The *x* axis represents the 16 time points and the *y* axis represents the total expression (TPM). *Nucleocytoviricota* family classifications are shown in brackets. Abbreviations: IM, *Imitervirales*; AG, *Algavirales*. (B) Eigengene of the genome coexpression modules. The *x* axis represents the 16 time points and the *y* axis represents the eigengene expression values (arbitrary units).

metatranscriptomes (39); in addition to recovering these genomes, we also identified transcripts mapping to an additional 39 metagenome-derived *Prasinoviridae* genomes, highlighting the diversity of viruses within this clade that were active in the same community during the sampling period. A wide range of other *Imitervirales* viruses were also present in addition to the *Mesomimiviridae* (Fig. 1c): *Mimiviridae* (IM_16), which includes the original *Acanthamoeba polyphaga* Mimivirus and its close relatives; IM_12 containing the recently cultivated Tetraselmis virus (TetV) (19); IM_08, which includes the recently reported Choanoflagellate virus (43); IM_09, which includes *Aureococcus anophagefferens* virus (AaV) (26); and *Imitervirales* families 7 and 13, which contain no cultivated representatives. One *Pimascovirales* virus (GVMAG-M-3300009467-15) and one *Asfuvirales* virus (GVMAG-M-3300027833-19) were also abundant in the transcriptomes, suggesting they may also infect microbial eukaryotes in the community. The *Asfuvirales* virus found here was also recently identified in waters off the coast of South Africa (7), indicating that it may be broadly distributed in the ocean. The presence of numerous divergent families of *Nucleocytoviricota* in the metatranscriptomes supports the view that marine systems harbor an immense diversity of these viruses that are actively infecting a wide variety of protists hosts.

We examined summed whole-genome transcriptional activity of the 145 *Nucleocytoviricota* to identify potential diel cycles of viral activity. We found 6 viruses that had significantly higher expression during nighttime periods (Fig. 3A, Mann-Whitney U test, corrected *P* value < 0.1). These transcriptional patterns were not strictly diel because their peak of expression was found at different times during the night, but they nonetheless suggest that viral activity may be higher at night for some of these viruses. We also

analyzed whole-genome transcriptional activity using a weighted network-based approach that groups viruses with similar temporal transcriptional profiles into modules. This analysis revealed two primary modules, one of which contained 60 viruses and had a module-wide transcriptional profile with notable peaks at nighttime periods (Module 1, Fig. 3B). We found no significant diel cycles in the *Nucleocytoviricota* whole-genome transcriptional profiles when we tested for strict diel periodicity (RAIN, *P* value < 0.1, see Materials and Methods), suggesting that although viral activity appears higher at night for some viruses, pronounced and significant diel cycling of viral activity was not detectable. This does not necessarily suggest that diel cycles are uncommon in marine *Nucleocytoviricota*; previous studies have found that the detection of diel cycles in metatranscriptomic data can require longer time-series (>20 time points) and may only be feasible for the most abundant community members for which gene expression is high (44).

The general trend of higher nighttime expression of *Nucleocytoviricota* is consistent with previous studies of marine viruses. A culture-based study of *Ostreococcus tauri* virus 5 found that viral transcripts increased markedly at night (45). Two of the *Nucleocytoviricota* with higher nighttime expression in our analysis belong to the *Prasinoviridae* (AG_01), which includes cultivated viruses that infect *Ostreococcus* and other prasinophytes, suggesting that higher nighttime activity may therefore be a common trait in this family. Previous work on bacteriophage has revealed peak viral transcription at various times throughout a 24-h period depending on the viral group (46–48), although in many cases gene expression was found to peak near dusk. It has been hypothesized that high viral expression at dusk is linked to the energetic state of phototrophic host cells such as *Prochlorococcus*, which grow throughout the day and divide at dusk. Viruses active near the end of the host growth period may therefore have more cellular resources to exploit for virion production (46). Whether or not a similar dynamic is at play with *Nucleocytoviricota* is unclear; the *Prasinoviridae* gene expression appears to peak after dusk, but this may be caused in part by a longer infection program of these viruses compared to cyanophages. The four *Mesomimiviridae* with diel transcriptional signatures that peak in the evening (IM_01, Fig. 3) may also infect phototrophic hosts where similar factors are at play. Many of the *Nucleocytoviricota* we detected in the transcriptomes likely infect heterotrophic hosts, however, and it remains unclear if there would be reason to expect a diel infection pattern in these viruses. Further work is therefore needed to examine the potential role of diel cycling in these viruses in more detail.

Throughout the sampling period, we detected expression of numerous viral genes involved in diverse metabolic processes (Fig. 4, Fig. S2, Data Set S2). As expected, we found high expression of viral core genes, including the major capsid protein, family B DNA polymerase, A32 packaging enzyme, VLTF3 transcriptional factor, and superfamily II helicase (Fig. 4); this is consistent with both culture-based and culture-independent studies of *Nucleocytoviricota* that have found most of these genes, and in particular the capsid proteins, to be highly expressed during infection (45, 49–52). Chaperones from the Hsp70 and Hsp90 families were also highly expressed in numerous *Nucleocytoviricota*, consistent with some previous transcriptome studies (50, 52), where these chaperones likely play a role in protein folding in virus factories, particularly for capsids (3, 52). We also detected transcripts involved in mitigating cellular stress, such as glutathione *S*-transferase (GST) and superoxide dismutase (SOD). SOD is involved in the detoxification of reactive oxygen species produced by hosts as a defense mechanism during infection, and it has been postulated that viral-encoded SODs may allow some viruses to infect a broader array of hosts (53). GST may be involved in the detoxification of electrophilic metabolites, and the host version of this gene was also found to be overexpressed at several time points of an infection experiment of *Aureococcus anophagefferens* virus and its host (54).

We also detected numerous transporters predicted to target nitrogen, sulfur, and phosphorus-containing nutrients, consistent with the view that *Nucleocytoviricota* manipulate the nutrient environment of their hosts during infection. Among these, we identified expression of several viral ammonium transporters; previous work with an

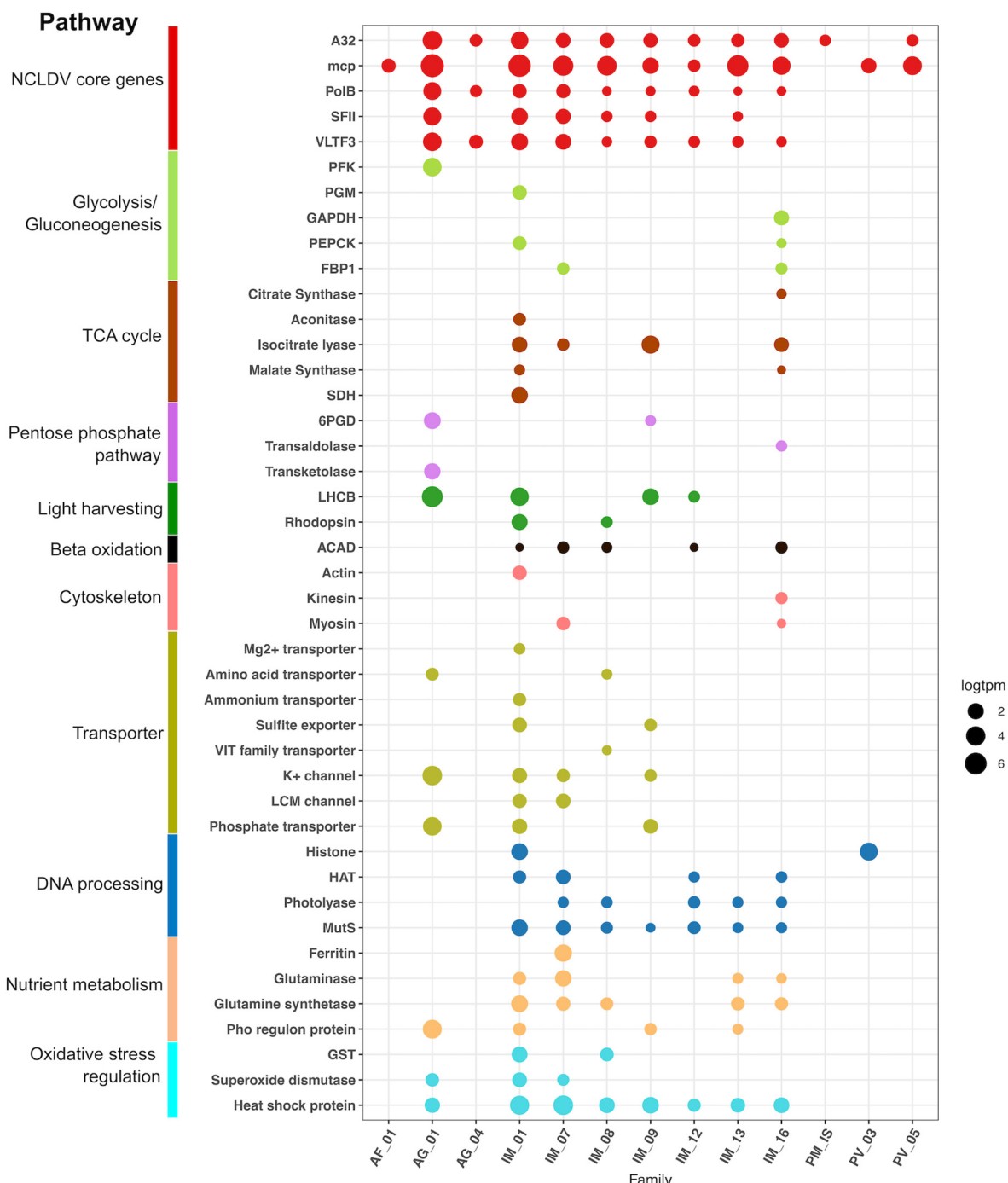

**FIG 4** Metabolic genes expressed in the transcriptomes. The *x* axis shows different viral clades and the *y* axis denotes the functional annotation. The sizes of the bubbles represent the total abundance of the gene in TPM (natural log transformed) and the colors show the functional category. Family assignment abbreviations: AF, *Asfuvirales*; IM, *Imitervirales*; AG, *Algavirales*; PM, *Pimascovirales*; PV, *Pandoravirales*; IS, *incertae sedis*. Gene function abbreviations: PFK, phosphofructokinase; PGM, phosphoglycerate mutase; GAPDH, glyceraldehyde 3-P dehydrogenase; PEPCK, phosphoenolpyruvate carboxykinase; FBP1, fructose-1-6-bisphosphatase; SDH, succinate dehydrogenase; 6PGD, 6-phosphogluconate dehydrogenase; LHCB, chlorophyll a/b binding protein; ACAD, acyl-CoA dehydrogenase; HAT, histone acetyltransferase; GST, glutathione *S*-transferase.

ammonium transporter encoded by *Ostreococcus tauri* virus 6 has shown that this gene is expressed during infection and manipulates nutrient uptake by the host cell (55). Viral-encoded phosphate transporters are also common in many marine *Nucleocytoviricota* (12, 56) and expression of these genes further emphasizes their likely role in viral-mediated nutrient transport during infection. Other genes involved in nutrient homeostasis were also well represented in the metatranscriptomes,

including ferritin, glutaminase, glutamine synthetase, and Pho regulon components (Fig. 4), consistent with recent findings that these genes are also common in these viruses (12).

Recent work has noted that many *Nucleocytoviricota* encode enzymes involved in central metabolic pathways, and it has been postulated that these have a role in the reprogramming of healthy cells into virocells during infection (12). We detected expression of many of these central metabolic enzymes throughout the sampling period; genes involved in glycolysis, the TCA cycle, the pentose phosphate pathway, and beta oxidation were all present in the metatranscriptomes, and most were consistently expressed across time points, especially in the *Imitervirales* (Fig. 4, Fig. S2). A recent study of TCA cycle enzymes encoded by a *Pandoravirus* isolate has suggested that these enzymes have activity consistent with bioinformatic predictions and that some are packaged into virions, and would therefore influence host physiology soon after initial infection (57). Together with our results, this suggests that viral manipulation of central metabolic processes is a widespread mechanism employed by many *Nucleocytoviricota*, and that this is a common occurrence in viral infections in the ocean. This raises the question of what proportion of the protist community is infected by *Nucleocytoviricota* at any given time, and thereby has an altered metabolic state. Previous studies examining infected hosts have found that 27 to 37% of cells can be infected by *Nucleocytovicota* during an algal bloom (58, 59); however, given the possibility that multiple viruses may infect the same host population, the total number of infected cells may be substantially higher. This leads to the surprising possibility that a large proportion of protist populations in some marine systems may consist of "virocells" in which their metabolism is substantially altered by viruses that infect them, at least for some periods of high viral activity.

Translation-related genes were once thought to be trademarks of cellular organisms until the discovery of their presence in viruses of the family *Mimiviridae*, and they have since become one of the most noteworthy features of the genomic repertoires of giant viruses. We observed high expressions of several aminoacyl-tRNA synthetase genes (aaRSs), including those that encode asparaginyl-tRNA synthetases, lysyl-tRNA synthetases, tyrosyl-tRNA synthetases, and prolyl-tRNA synthetases. Transcripts mapping to these genes were found from *Imitervirales* genomes in the *Mimiviridae*, *Mesomimiviridae*, IM_07, IM_12, and IM_13 families. Proteins encoded by the aaRS genes are responsible for the interaction between tRNAs and their amino acids and have been previously found in diverse members of the *Mimiviridae* (60, 61), and may be associated with an increase of viral fitness (62). We also detected viral transcripts from *Imitervirales* genes with homology to genes involved in translation initiation, such as translation initiation factor 4E (IF4E), translation initiation factor SUI1, and translation initiation factor 3 (IF3). Translation elongation factors EF-TU, EIF-5a, and EF-P were also expressed, especially in the *Mimiviridae* and families IM_07, IM_09, and IM_13. This is consistent with the finding that many mimivirus genomes encode various translation factors (63, 64). The expression of viral genes involved in translation suggests a degree of independence from the host's translational machinery during active infections occurring in oceanic waters.

Surprisingly, we also found several transcripts matching to predicted cytoskeletal proteins, including actin, myosin, and kinesin. Recent studies have identified divergent actin, myosin, and kinesin homologs in *Nucleocytoviricota* and, in some cases, it has been suggested they may have played a role in the emergence of this protein in extant eukaryotic lineages (65–67). We found three expressed viral genes with predicted myosin motor domains and one with kinesin domains, and subsequent analysis of all *Nucleocytoviricota* genomes in our reference database recovered an additional 109 myosin and 200 kinesin homologs. To examine if these genes were recently acquired from cellular hosts, we performed a phylogenetic analysis of the viral myosin and kinesin proteins with references; our results indicate that the viral proteins typically form separate branches that are distinct from their eukaryotic homologs, except in a few cases where it appears some *Nucleocytoviricota* have acquired cellular copies more recently

mSystems®

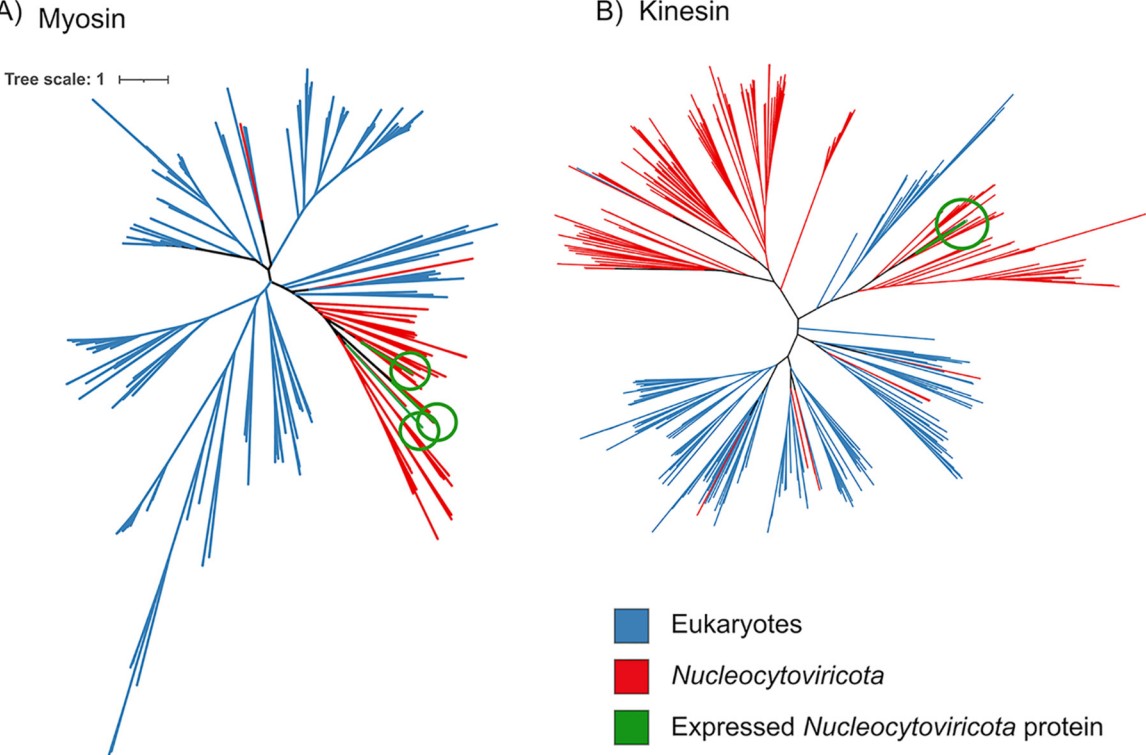

**FIG 5** Phylogeny of *Nucleocytoviricota* myosin (left) and kinesin (right) proteins together with references available in the EggNOG database. The myosin and kinesin genes that were identified in the metatranscriptomes are colored green.

(Fig. 5). This is consistent with recent studies revealing a dynamic gene exchange between *Nucleocytoviricota* and their hosts that in some cases dates back to the early diversification of eukaryotes (16, 68–71). The expression of viral genes with homology to actin, myosin, and kinesin domains suggests that these viruses may use these proteins to manipulate host cytoskeletal dynamics during infection. It is thus possible that these proteins play a role in the establishment and maintenance of perinuclear viral factories, which are responsible for viral morphogenesis or for the subcellular trafficking of viral particles. In *Aureococcus anophagefferens*, cytoskeletal genes are downregulated during infection by its virus AaV (54); if this is common across other host-virus systems, it would suggest that *Nucleocytoviricota*-encoded myosin, kinesin, and actin proteins may help to maintain the functioning of the cytoskeleton during later periods of infection.

Recently it has been shown that giant viruses commonly endogenize into the genomes of their hosts, which raises the possibility that some of the transcriptional patterns we identified in this study are due to giant endogenous viral elements (GEVEs) rather than free viruses (68). Previous examination of the transcriptional landscape of GEVEs indicated that the hallmark *Nucleocytoviricota* genes involved in information processing, such as major capsid protein and DNA polymerase, are typically not expressed. While numerous genes of GEVEs show various levels of expression, others lack detectable expression under normal growth conditions (68), contrary to patterns observed during the active viral infection (50, 52, 55). The patterns we report here for the metatranscriptomes are therefore more consistent with the gene expression patterns of viruses undergoing active viral replication than the transcriptional activity of GEVEs. Nonetheless, the presence of GEVEs in many algal genomes raises the possibility that not all transcription of viral genes in a microbial community can be directly linked to viral propagation.

To potentially link viruses to their hosts, we performed a network-based analysis based on correlations of whole-genome transcription of both viruses and eukaryotic

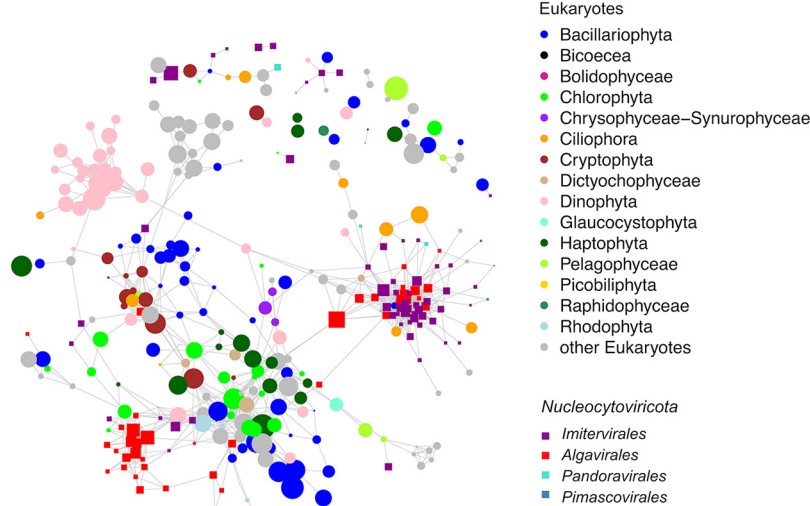

**Eukaryotes**
- Bacillariophyta
- Bicoecea
- Bolidophyceae
- Chlorophyta
- Chrysophyceae–Synurophyceae
- Ciliophora
- Cryptophyta
- Dictyochophyceae
- Dinophyta
- Glaucocystophyta
- Haptophyta
- Pelagophyceae
- Picobiliphyta
- Raphidophyceae
- Rhodophyta
- other Eukaryotes

*Nucleocytoviricota*
- *Imitervirales*
- *Algavirales*
- *Pandoravirales*
- *Pimascovirales*

**FIG 6** Host-virus expression correlation network. Each node represents a eukaryotic genome (circle) or viral genome (square). The lines between two genomes denote a correlation coefficient larger than 0.8. The sizes of the nodes represent the total abundance in TPM (natural log transformed) of the genome and the colors show the genome's taxonomy.

plankton (Fig. 6). This was motivated by several studies that have noted how the transcriptional activity of many viruses is often correlated with that of their host *in situ* (39, 49, 72). The resulting network links *Nucleocytoviricota* to diverse potential hosts, including Dinophyta, Ciliophora, Stramenopiles, and Pelagophyta. Many members of the *Prasinoviridae* clustered in the same part of the network as *Ostreococcus*, consistent with the prediction that most members of this family are Prasinoviruses. Several viruses of the family *Mesomimiviridae* were correlated with members of the diatom genera *Leptocylindrus*, *Chaeotoceros*, and *Eucampia*, consistent with a recent co-occurrence analysis from Tara Oceans data which suggested that some *Nucleocytoviricota* are potentially associated with diatoms (73); however, this association remains speculative pending more detailed analysis. In general, these results support the view that *Nucleocytoviricota* likely infect diverse hosts in the environment, but specific host predictions should be treated with caution pending validation with experimental approaches.

In conclusion, in this study we identified 145 *Nucleocytoviricota* with high transcriptional activity over a 60-h period in surface waters of a coastal marine system. We analyzed only genomes for which >10% of the predicted genes could be recovered at some point during the time series; this provides high confidence that the detected viruses were indeed present, but it also implies that the actual number of active viruses is far higher than 145 because many viral transcripts remain below the level of detection. This therefore further highlights the remarkable diversity of active *Nucleocytoviricota* in this coastal system and implies more generally that these viruses are an important component of global marine environments. In addition, we found expression of diverse viral-encoded functional genes from central metabolic pathways, as well as genes with similarity to eukaryotic cytoskeletal components, emphasizing the complex mechanisms employed by giant viruses to manipulate their hosts in the environment. One implication of these findings is that these viruses also have a potentially important role in shaping marine food webs; the relative impact of top-down versus bottom-up controls in marine systems has long been debated (74), and the ability of *Nucleocytoviricota* to infect both heterotrophic protists (i.e., grazers) and phototrophic plankton (i.e., primary producers) suggests they can influence the relative impact of these forces. Future work identifying the hosts of these viruses and their relative contribution to eukaryotic plankton mortality rates will therefore be important avenues of future work. Overall, our findings suggest that

*Nucleocytoviricota* are an underappreciated component of marine systems that exert an important influence on community dynamics and biogeochemical cycling.

## MATERIALS AND METHODS

**Compilation of a *Nucleocytoviricota* genome database for read mapping.** We compiled a set of *Nucleocytoviricota* genomes for transcript mapping that included metagenome-assembled genomes (MAGs) as well as genomes of cultured isolates. For this, we first downloaded 2,074 genomes from Schulz et al. (70) and 501 genomes from Moniruzzaman et al. (12), two recent studies which generated numerous *Nucleocytoviricota* MAGs. We also included all *Nucleocytoviricota* genomes available in NCBI RefSeq as of 1 June 2020. Last, we also included several *Nucleocytoviricota* genomes from select publications that were not yet available in NCBI, such as the cPacV, ChoanoV, and AbALV viruses that have recently been described (43, 50, 75).

After compiling this set, we dereplicated the genomes, since the presence of highly similar or identical genomes can obfuscate the results of read mapping. For dereplication, we compared all *Nucleocytoviricota* genomes against each other using MASH v. 2.0 (76) ("mash dist" parameters -k 16 and -s 300), and clustered genomes together using a single-linkage clustering, with all genomes with a MASH distance of ≤0.05 linked together. The MASH distance of 0.05 was chosen since it has been found to correspond to an average nucleotide identity (ANI) of 95% (76), which has further been suggested to be a useful threshold for distinguishing viral species-like clusters (77). From each cluster, we chose the genome with the highest $N_{50}$ contig length as the representative. Prior to read mapping, we also decontaminated the genomes through analysis with ViralRecall v.2.o (78) (-c parameter), with all contigs with negative scores removed. The final database used for read mapping contained 2,431 viral genomes.

Our read mapping strategy employed a translated homology search (DNA searched against protein), and so we predicted proteins from all genomes using Prodigal 2.6.3 (79) (run independently on each genome using default parameters), and then masked the protein database using tantan v. 22 (80) (default parameters) to prevent spurious read mapping to low-complexity sequences. We then formatted the database for read mapping using the lastdb utility in LAST v. 1060 (81).

**Metatranscriptome data set.** We examined the metatranscriptomic data set of 16 time points obtained from the >5-$\mu$m size fraction microbial communities previously reported for the California Current system (39). In this study, community RNA samples were collected every 4 h over a 60-h time frame, from 6 a.m. 16 September to 6 p.m. 18 September. Details regarding sample processing have been previously described (39, 82). We downloaded and trimmed reads from each of the 16 metatranscriptome libraries with Trim Galore v. 0.6.4 with parameters "–length 50 -q 5 –stringency 1" and, after removing singleton reads, we then mapped paired reads onto the *Nucleocytoviricota* database using LASTAL v. 1060 (81) (parameters -m 10 -u 2 -Q 1 -F 15). We only considered hits with percent identity of >95% and bit score of >50. This workflow is based on methods that have been successfully employed in previous studies (44, 46). For relative abundance comparisons, read mapping counts were converted to transcripts per kilobase per million (TPM) (83). To avoid the spurious detection of viral genomes in the metatranscriptomes, we only considered genomes for which ≥10% of the proteins in that genome had hits; this cutoff was guided by recent work which has suggested that read mapping to >10% of a genome are unlikely to be due to spurious mappings, and therefore are indicative of the presence of that virus in a given sample (84). After performing this filtering, we arrived at a set of 145 *Nucleocytoviricota* reference genomes that were used in subsequent analysis.

**Protein annotations.** We annotated proteins in our *Nucleocytoviricota* database by comparing them to the EggNOG 4.5 (85) and Pfam v. 31 (86) hidden Markov models (HMMs) using HMMER3 3.3 (parameter "-E 1e-3" for the NOG search and "–cut_nc" for the Pfam search). In both cases only best hits were retained. These annotations are available in Data Set S2 in the supplemental material. Annotations were manually reviewed to examine functions of interest, including energy metabolism, carbohydrate metabolism, cytoskeleton, membrane transport, light harvesting, and oxidative stress regulation. Hallmark *Nucleocytoviricota* proteins were identified using HMMs previously described (12).

**Nucleocytoviricota phylogeny and clade delineation.** To provide phylogenetic context for the 145 genomes we identified in the metatranscriptomes, and to delineate broader clades, we constructed a large multilocus phylogenetic tree of *Nucleocytoviricota* using the marker genes DNA polymerase B (PolB), A32 packaging enzyme (A32), superfamily II helicase (SFII), VLTF3 transcription factor (VLTF3), RNA polymerase large subunit (RNAPL), topoisomerase family II (TopoII), and transcription factor IIB (TFIIB). We previously benchmarked this marker set and found it is suitable for making phylogenetic trees across all 6 *Nucleocytoviricota* orders (5). For this tree, we used all 145 genomes present in the metatranscriptomes as well as select reference *Nucleocytoviricota* sampled from across all of the main orders (*Asfuvirales*, *Chitovirales*, *Pimascovirales*, *Algavirales*, *Pandoravirales*, and *Imitervirales*). The concatenated phylogeny was generated with the ncldv_markersearch.py script, with default parameters (github.com/faylward/ncldv_markersearch), the alignment was trimmed with TrimAl v. 1.4.rev22 (87) (parameter -gt 0.1), and the tree was inferred using IQ-TREE (88) with the LG+F+I+G4 model. Support values were inferred using 1,000 ultrafast bootstraps (89). We previously delineated family-level clades for the *Nucleocytoviricota* (5) and we used this nomenclature to do the same for this tree.

**Virus-host transcriptional network.** To identify potential marine host-virus relationships, we computed a Pearson correlation matrix between the total abundance of all genomes with gene expression across the 16 time points (function cor() in R). The total abundance of genomes was calculated as the summed abundance of all genes of the genome expressed. Host transcriptional abundances were obtained from Kolody et al (39). The abundance of individual genes was calculated by normalizing the

number of raw reads by the total number of raw reads per million in each sample. Only correlation coefficients between genome pairs exceeding 0.8 were reported.

**Tests for diel cycling and day/night overexpression.** To explore the transcriptional activities and possible diel expression patterns of the *Nucleocytoviricota* community, we examined the TPM-normalized transcriptional profiles of individual *Nucleocytoviricota* genes using RAIN (90). We did the same for each of the 145 *Nucleocytoviricota* genomes present in the transcriptomes by summing TPM expression values across each genome. *P* values for these tests were corrected using the Benjamini and Hochberg method (91), as implemented by the p.adjust command in R.

In addition to strict tests for diel periodicity, we also sought to examine potential overexpression of *Nucleocytoviricota* in daytime versus nighttime periods. For this we performed a Mann-Whitney U test on the summed gene expression of each virus, using time points of 0600, 1000, and 1400 as daytime and time points of 1800, 2200, and 0200 as nighttime (wilcox.test in R). For these tests, we also corrected the *P* values using the same method as the RAIN tests.

To find clusters (modules) of genomes with highly correlated temporal expression, we performed the weighted gene coexpression network analysis (WGCNA) method (92) on the TPM-normalized transcriptional profiles of the *Nucleocytoviricota* genome-wide transcriptional profiles. We used a soft thresholding power of 14, parameters TOMType ="signed," reassignThreshold = 0, minModuleSize = 10, and mergeCutHeight = 0.25. We then used the moduleEigengenes utility in the WGCNA package to summarize the genome module's average expression profile (module eigengene).

**Phylogeny of myosin and kinesin proteins.** To examine the evolutionary history of the *Nucleocytoviricota* proteins with myosin and kinesin domains, we identified all occurrences of these proteins in our viral genome database, including those proteins that were not expressed in the metatranscriptomes. For this analysis, we considered proteins with matches to the PF00063 (myosin) and PF00225 (kinesin) (with only scores greater than the noise cutoff considered). For references, we downloaded all the kinesin (COG5059) and myosin (COG5022) homologs from the EggNOG database v4.5. Many of the species harbor multiple copies of these proteins and thus, for the purpose of broad phylogenetic analysis, we randomly selected one copy of these genes per species. Diagnostic phylogenetic trees were constructed using FastTree v. 2 (93) implemented in ete3 toolkit (94) for the purpose of removing long branches and redundant or small sequences (<300 amino acids long). To construct the final trees, we used ClustalOmega to align the sequences and trimAl (parameter -gt 0.1) to trim the alignments. We used IQ-TREE to build maximum likelihood phylogenies with the model LG+I+G4 and assessed the node support with 1,000 ultrafast bootstrap replicates.

**Data availability.** The data sets analyzed in this study are already publicly available and were accessed as described in the Materials and Methods section.

## SUPPLEMENTAL MATERIAL

Supplemental material is available online only.
**DATA SET S1**, XLSX file, 0.03 MB.
**DATA SET S2**, XLSX file, 1 MB.
**FIG S1**, TIF file, 2.7 MB.
**FIG S2**, TIF file, 2.6 MB.

## ACKNOWLEDGMENTS

We acknowledge the use of the Virginia Tech Advanced Research Computing Center for bioinformatic analyses performed in this study.

We are thankful to the members of the Aylward Lab and Andrew Allen for their help with a previous version of the manuscript.

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
