## [Reviewer comments · mSystems]

High Transcriptional Activity and Diverse Functional Repertoires of Hundreds of Giant Viruses in a Coastal Marine System

Anh Ha, Mohammad Moniruzzaman, and Frank Aylward

Corresponding Author(s): Frank Aylward, Virginia Tech

Review Timeline:

Submission Date:	March 10, 2021
Editorial Decision:	April 18, 2021
Revision Received:	May 24, 2021
Accepted:	May 26, 2021

Editor: Seth Bordenstein

Reviewer(s): Disclosure of reviewer identity is with reference to reviewer comments included in decision letter(s). The following individuals involved in review of your submission have agreed to reveal their identity: James L. Van Etten (Reviewer #1)

Transaction Report:

DOI: <https://doi.org/10.1128/mSystems.00293-21>

April 18, 2021

Prof. Frank O Aylward
Virginia Tech
Blacksburg

Re: mSystems00293-21 (High Transcriptional Activity and Diverse Functional Repertoires of Hundreds of Giant Viruses in a Coastal Marine System)

Dear Prof. Frank O Aylward:

Thank you for submitting your manuscript to mSystems. We have completed our review, and I am pleased to inform you that, in principle, I expect to editorially accept it for publication in mSystems upon minor revisions.

Both reviewers and I thought this was a nicely done study. In addition to structural gene expression, your discoveries of expression of viral genes involved in cellular stress regulation, carbon metabolism, and possible manipulation of host cytoskeleton dynamics will be of significant interest, especially the unusual expression of viral-encoded genes with annotated domains to actin, myosin, and kinesin.

Thank you for the privilege of reviewing your work. Below you will find instructions from the mSystemseitorial office and comments generated during the review.

Preparing Revision Guidelines

For complete guidelines on revision requirements, please see the Instructions to Authors at [link to page]. **Submissions of a paper that does not conform to mSystems guidelines will delay acceptance of your manuscript.**

Due to the SARS-CoV-2 pandemic, our typical 60 day deadline for revisions will not be applied. I hope that you will be able to submit a revised manuscript soon, but want to reassure you that the journal will be flexible in terms of timing, particularly if experimental revisions are needed. When you are ready to resubmit, please know that our staff and Editors are working remotely and handling

submissions without delay. If you do not wish to modify the manuscript and prefer to submit it to another journal, please notify me of your decision immediately so that the manuscript may be formally withdrawn from consideration by mSystems.

Sincerely,

Seth Bordenstein

Editor, mSystems

Journals Department
Reviewer comments:

Reviewer #1 (Comments for the Author):

This manuscript describes a large transcriptome analysis of giant viruses (phylum Nucleocytoviricota) from a coastal marine system. They identify 145 viral transcriptomes, which probably only represents a small part of the entire population of giant viruses. Not surprisingly they find viral genes being expressed that encode an amazing number of unexpected proteins.

I have two general comments. First the references are a mess. Some are missing the vol and page numbers of the articles - e.g. ref 2,4,5 etc. Others are missing the author's names, e.g., 6. Some are missing the title of the journal and some have the first word in the title capitalized and others do not.

Second, This same lab recently published a beautiful paper where they showed that some algae had giant virus genomes embedded in their genomes. Therefore, one would think that expression of some of these algal genomes might contain viral transcripts that would be included in the current transcription analysis. There is nothing wrong with this but I am surprised that this was not even mentioned in the paper.

Reviewer #2 (Comments for the Author):

Anh D Ha, Mohammad Moniruzzaman and Frank O Aylward present data on giant viruses transcriptional activity of a coastal marine system. Authors described data on gene expression in a context of 2.5 day time course. The data is a good contribution to the understanding of giant viruses metabolic and ecological dynamics in such system.

Major points:

- Would be important authors provide to the readership detailed information regarding cDNA preparation for this study. I am aware that authors cited in the methods a previous study that describe it, but such cited study cited another paper... Were the samples treated with DNase before cDNA preparation? If not, how the authors differentiate genomic and transcript sequences? It is a simple and important information.
- Translation related genes represent an important hallmark on Mimiviridae genomes (maybe the most important). Would be useful authors provide detailed information about those genes, e.g. aminoacyl tRNA synthetases. What about tRNAs?
- Regarding the virosphere, to my knowledge, krebs cycle gene has been first described in tupanvirus - citrate synthase (<https://www.sciencedirect.com/science/article/pii/S0065352718300447>)
- It is crystal-clear the importance of metagenomics to access viral diversity. But the term "cultivated or non-cultivated viruses" can lead to misinterpretation. New isolation systems can be developed and quickly transform non-cultivated to cultivated viruses.
- The day / night expression analyses are very fine. Although it has already described for *Ostreococcus tauri* virus, it deserves a more in depth discussion on its ecological and metabolic causes and consequences.

Response to Reviewer Comments

Brief note on the resubmission:

We have recently published a taxonomy of *Nucleocytoviricota* and delineated numerous novel families within this viral phylum (Aylward et al., BioRxiv, 2021). We have therefore revised the nomenclature of this manuscript so that it is consistent with our other study. We feel this is important for consistency, and to avoid confusion stemming from the use of different nomenclature in different studies, but this does not change any of the conclusions of the current study.

This manuscript describes a large transcriptome analysis of giant viruses (phylum Nucleocytoviricota) from a coastal marine system. They identify 145 viral transcriptomes, which probably only represents a small part of the entire population of giant viruses. Not surprisingly they find viral genes being expressed that encode an amazing number of unexpected proteins.

I have two general comments. First the references are a mess. Some are missing the vol and page numbers of the articles - e.g. ref 2,4,5 etc. Others are missing the author's names, e.g., 6. Some are missing the title of the journal and some have the first word in the title capitalized and others do not.

Thank you for pointing this out. We experienced a problem with the citation manager that we used, which created several errors we did not catch when we formatted the final draft for submission. We have corrected these issues, and we have also re-uploaded the manuscript to BioRxiv so that the work we cite is appropriately acknowledged in this venue as well.

Second, This same lab recently published a beautiful paper where they showed that some algae had giant virus genomes embedded in their genomes. Therefore, one would think that expression of some of these algal genomes might contain viral transcripts that would be included in the current transcription analysis. There is nothing wrong with this but I am surprised that this was not even mentioned in the paper.

Thank you very much for your kind words about our recent study in *Nature* (Moniruzzaman et al., 2020). In this study we found numerous Giant Endogenous Viral Elements (GEVEs) in the genomes of green algae, and there is evidence that other protists hosts also harbor similar viral elements. Although we did find expression of GEVE genes in our previous study, it was typically quite low compared to host genes, and core viral genes were usually not detected (MCP, PoIB, other structural proteins, etc). So although GEVEs play an important role in eukaryotic genome evolution, and their presence may impact the host transcriptional landscape, current data suggests transcriptional from GEVE is quite a bit lower than that of viruses undergoing active infection. In our present study we found high expression of core *Nucleocytoviricota* genes in the metatranscriptomes, and we only examined viral genomes for which >10%

of their encoded genes could be identified across all samples. We therefore believe it is unlikely that the results we report are due to GEVEs, and that the patterns are more consistent with active free viruses.

These points notwithstanding, we must acknowledge that our understanding of GEVEs is in its infancy, and we must agree with this reviewer that we cannot completely rule out the possibility that the transcription of GEVE genes could contribute to the overall patterns we report. We have therefore added a discussion of these points into the manuscript (lines 286-297).

Reviewer #2 (Comments for the Author):

Anh D Ha, Mohammad Moniruzzaman and Frank O Aylward present data on giant viruses transcriptional activity of a coastal marine system. Authors described data on gene expression in a context of 2.5 day time course. The data is a good contribution to the understanding of giant viruses metabolic and ecological dynamics in such system.

Major points:

- Would be important authors provide to the readership detailed information regarding cDNA preparation for this study. I am aware that authors cited in the methods a previous study that describe it, but such cited study **cited another paper... Were the samples treated with DNase before cDNA preparation? If not, how the authors differentiate genomic and transcript sequences?** It is a simple and important information.

Thank you very much for pointing this out. Detailed methods are provided in Shi et al., (Nature, 2009), including the DNase procedure. We now cite this study. Because the dataset we use is public we do not feel that a detailed description of the methods used to generate it are appropriate, but by including this citation we feel it will be easier for readers to trace the methods back.

- Translation related genes represent an important hallmark on Mimiviridae genomes (maybe the most important). Would be useful authors provide detailed information about those genes, e.g. aminoacyl tRNA synthetases. What about tRNAs?

We agree that a more detailed discussion of translation-associated genes is warranted, given the importance of these genes for *Mimiviridae*. We now provide a dedicated paragraph to the discussion of viral-encoded translation genes we identified, together with additional references to provide proper context (lines 245-261).

- Regarding the virosphere, to my knowledge, krebs cycle gene has been first described in tupanvirus - citrate synthase
(<https://www.sciencedirect.com/science/article/pii/S0065352718300447>)

Thank you for pointing this out. We agree this is an important contribution that should be cited when we first refer to the presence of TCA cycle enzymes in giant viruses (line 66). We will make sure to refer to this study in our future work as well.

- It is crystal-clear the importance of metagenomics to access viral diversity. But the term "cultivated or non-cultivated viruses" can lead to misinterpretation. New isolation systems can be developed and quickly transform non-cultivated to cultivated viruses.

Thank you for pointing this out. We do not wish to imply that some groups cannot be cultivated. In fact, we are confident (and hopeful) that many of these viral lineages will be cultivated in the future as methods improve and more labs begin to examine these exciting viruses. We only mention this because readers may be interested to know how many of the genomes we analyze in detail are from cultivated representatives, and how many were produced using a cultivation-independent approach (metagenomics, single cell sequencing, etc). We have revised the wording and simplified this sentence to read "... genomes derived from cultivation-independent methods" (line 114).

The term cultivation-independent is appropriate here because methods range from metagenome-based binning to single-cell based sequencing. Rather than getting into the details of the multiple studies that produced these genomes, we simply use the general term "cultivation-independent", similar to how the term is often used for bacteria and archaea.

- The day / night expression analyses are very fine. Although it has already been described for *Ostreococcus tauri* virus, it deserves a more in depth discussion on its ecological and metabolic causes and consequences.

We agree, and we have expanded the discussion of diel cycles (lines 179-193).

May 26, 2021

Prof. Frank O Aylward
Virginia Tech
Blacksburg

Re: mSystems00293-21R1 (High Transcriptional Activity and Diverse Functional Repertoires of Hundreds of Giant Viruses in a Coastal Marine System)

Dear Prof. Frank O Aylward:

Thank you and the coauthors for the revision. Your manuscript has been editorially accepted, and I am forwarding it to the ASM Journals Department for publication. For your reference, ASM Journals' address is given below. Before it can be scheduled for publication, your manuscript will be checked by the mSystems senior production editor, Ellie Ghatineh, to make sure that all elements meet the technical requirements for publication. She will contact you if anything needs to be revised before copyediting and production can begin. Otherwise, you will be notified when your proofs are ready to be viewed.

We recognize that the video files can become quite large, and so to avoid quality loss ASM suggests sending the video file via <https://www.wetransfer.com/>. When you have a final version of the video and the still ready to share, please send it to Ellie Ghatineh at eghatineh@asmusa.org.

Sincerely,

Seth Bordenstein
Editor, mSystems

Journals Department
Supplemental Data S2: Accept
Fig. S1: Accept
Supplemental Data S1: Accept
Fig. S2: Accept